# Clinical Heterogeneity of Early-Onset Autoimmune Gastritis: From the Evidence to a Pediatric Tailored Algorithm

**DOI:** 10.3390/diseases13050133

**Published:** 2025-04-25

**Authors:** Ivan Taietti, Martina Votto, Riccardo Castagnoli, Mirko Bertozzi, Maria De Filippo, Antonio Di Sabatino, Ombretta Luinetti, Alessandro Raffaele, Alessandro Vanoli, Marco Vincenzo Lenti, Gian Luigi Marseglia, Amelia Licari

**Affiliations:** 1Department of Clinical, Surgical, Diagnostic, and Pediatric Sciences, University of Pavia, 27100 Pavia, Italy; ivan.taietti@gmail.com (I.T.); riccardo.castagnoli@unipv.it (R.C.); mirko.bertozzi@unipv.it (M.B.); maria.defilippo01@universitadipavia.it (M.D.F.); gianluigi.marseglia@unipv.it (G.L.M.); amelia.licari@unipv.it (A.L.); 2Pediatric Clinic, Fondazione IRCCS Policlinico San Matteo, 27100 Pavia, Italy; 3Pediatric Surgery Unit, Fondazione IRCCS Policlinico San Matteo, 27100 Pavia, Italy; a.raffaele@smatteo.pv.it; 4Department of Internal Medicine and Medical Therapeutics, University of Pavia, 27100 Pavia, Italy; antonio.disabatino@unipv.it (A.D.S.); marco.lenti@unipv.it (M.V.L.); 5First Department of Internal Medicine, Fondazione IRCCS Policlinico San Matteo, 27100 Pavia, Italy; 6Unit of Anatomic Pathology, Fondazione IRCCS Policlinico San Matteo, 27100 Pavia, Italy; o.luinetti@smatteo.pv.it (O.L.); alessandro.vanoli@unipv.it (A.V.); 7Department of Molecular Medicine, Unit of Anatomic Pathology, University of Pavia, 27100 Pavia, Italy

**Keywords:** autoimmune gastritis (AIG), anti-parietal cell antibodies (APCA), children

## Abstract

Autoimmune gastritis (AIG) is an uncommon and often underestimated condition in children, characterized by chronic stomach inflammation leading to the destruction of oxyntic glands with subsequent atrophic and metaplastic changes. This condition is associated with hypo-/achlorhydria, impairing iron and vitamin B12 absorption. The pathogenesis involves the activation of helper type 1 CD4+/CD25-T-cells against parietal cells. Clinical manifestations in children are not specific and include abdominal pain, bloating, nausea, vomiting, and iron deficiency anemia (IDA). The disease is also linked to an increased risk of pernicious anemia, intestinal-type gastric cancer, and type I neuroendocrine tumors. AIG is often diagnosed through the presence of autoantibodies in the serum, such as parietal cell (APCA) and intrinsic factor (IF) antibodies. However, therapeutic recommendations for pediatric AIG are currently lacking. We aim to present two clinical cases of pediatric-onset AIG, highlighting the heterogeneous clinical manifestations and the challenges in diagnosis with the support of an updated literature review. A 9-year-old girl presented with refractory IDA, initial hypogammaglobulinemia, and a 12-year-old boy was initially diagnosed with eosinophilic esophagitis. Both cases underline the importance of considering AIG in children with chronic gastrointestinal symptoms and gastric atrophy. Diagnostic workup, including endoscopy and serological tests, is crucial for accurate identification. A better understanding of this condition is imperative for timely intervention and regular monitoring, given the potential long-term complications, including the risk of malignancy. These cases contribute to expanding the clinical spectrum of pediatric AIG and highlight the necessity for comprehensive evaluation and management in affected children.

## 1. Introduction

Autoimmune gastritis (AIG) is a rare and often underrecognized condition in children [1]. AIG is a chronic inflammatory disorder of the stomach, which is typically restricted to the corpus, characterized by the destruction of oxyntic glands and their replacement by atrophic and metaplastic mucosa, and usually accompanied by lymphoplasmacytic infiltration of the lamina propria and the gradual destruction of parietal cells [2]. The pathophysiological consequence of this phenomenon is hypochlorhydria or achlorhydria, which interferes with intestinal iron absorption and loss of intrinsic factor (IF), which compromises vitamin B12 intestinal absorption [3]. The exact pathogenetic mechanism is not fully understood [4]. Still, it is characterized, though inconstantly, by the presence in the serum of autoantibodies against the proton pump H^+^/K^+^ adenosine triphosphatase of gastric parietal cells (anti-parietal-cells antibodies [APCA]) and IF [5,6,7,8]. The etiopathogenetic process involves the activation of T helper type 1 CD4^+^/CD25^−^ T-cells directed against parietal cells [9,10]. Macroscopically, there is thinning or flattening of the gastric mucosa *rugae*. Histologically, there is a loss of gastric glandular structures in the oxyntic mucosa, inappropriately replaced by inflammatory cells and metaplastic epithelium or fibrous tissue [9,10]. All these features result in achlorhydria and hypergastrinemia, with the consequent proliferation of enterochromaffin-like cells (ECL) [9,10,11].

AIG is a well-known cause of pernicious anemia ([PA] megaloblastic anemia and vitamin B12 deficiency) in adult patients. Children generally show nonspecific and heterogeneous symptoms, including abdominal pain, nausea, vomiting, and loss of appetite, that are not responsive to treatments [2,3,4,5]. However, the most common clinical manifestation in children is iron deficiency anemia (IDA), consistent with the more rapid depletion of iron storage compared to vitamin B12 [2,9,12]. Diagnosis often involves a combination of medical history, physical examination, blood tests to check for anemia and vitamin deficiencies, and endoscopy with biopsy to examine the stomach lining. Treatment generally involves addressing nutrient deficiencies, mainly iron deficiency in children, and, where needed, vitamin B12 supplements. AIG may require ongoing monitoring and management, as it can affect a child’s growth and development [13]. Intestinal-type gastric cancer and type I neuroendocrine tumors are two possible oncological consequences of AIG [14]. Considering children’s long-life expectancy and the cancer risk, endoscopic surveillance of AIG every five years has been suggested [14].

Although a variety of therapeutic approaches have been suggested, including netazepide, somatostatin, or antrectomy to reduce the amount of circulating gastrin, to date, no treatment recommendations exist for pediatric patients with AIG, regardless of the absence or presence of metaplastic change [15].

Considering the rarity of the disease in children and the lack of age-specific guidelines, we aim to present two cases of pediatric-onset AIG, discuss their features and diagnostic-therapeutic work-up, and compare our experience with the currently available literature. A better understanding of AIG is mandatory, particularly regarding the nonspecific and heterogeneous clinical and histopathological features in children and adolescents. An early and precise diagnosis is crucial to better manage these patients for two reasons: the first is that a correct nutritional state is fundamental for proper development; the second is that the principal and worst complication is the risk of gastric neoplastic diseases. Thus, regular endoscopic follow-up could impact the prognosis of these patients.

## 2. Methods

We retrospectively enrolled two children diagnosed with AIG who have been followed at our Pediatric Clinic in Pavia, Italy. Diagnosis of AIG was made according to current Italian guidelines and is based on clinical, serological, and histological features [15]. Clinical, endoscopic, and histological data were collected during diagnosis and follow-up visits. Findings from radiological examinations were also reported when performed. Patients were screened for several autoimmune diseases, including autoimmune thyroiditis (AITD), celiac disease (CeD), type 1 diabetes mellitus (T1DM), and inflammatory bowel diseases (IBDs). We also ruled out immunodeficiencies. We included patients whose parents signed written informed consent. The Ethical Committee of Fondazione IRCCS San Matteo di Pavia, Italy, approved this study (protocol number: 0003241/22).

The literature review was performed using the online database PubMed and the MeSH terms “autoimmune gastritis” OR “atrophic gastritis” AND “children.” The research was conducted in November 2024. We included retrospective studies, cross-sectional and cohort studies, case series, and case reports published in English and peer-reviewed journals in which participants were children and adolescents (0–18 years) with a diagnosis of histologically confirmed AIG, without limits on the year of publication. Two authors manually screened and reviewed potentially eligible publications and excluded non-relevant publications (e.g., narrative/systematic review, languages other than English, adult population, and other pathological conditions).

## 3. Clinical Case Description

### 3.1. Case 1

A 9-year-old girl was referred to our Pediatric Clinic for unexplained asthenia and weight loss during the previous year. Her family history was negative for chronic or infectious diseases. Her grandfather died because of an unspecified gastric neoplasia. The first clinical observation and physical examination were negative. The initial diagnostic assessment showed severe iron deficiency anemia (IDA) (hemoglobin 7 g/dL), treated with oral iron therapy, and IgG deficiency with mild IgG1 and IgG2 subclasses deficiency (Table 1). Lymphocyte subsets showed a mild reduction in B-lymphocytes. Vaccine immune responses (hepatitis B, measles, tetanus, and diphtheria) were conserved. An in vitro proliferation test (response to mitogens and IgG production) resulted within the normal range. Comprehensive autoimmune screening was negative. *Helicobacter pylori* (*H. pylori*) fecal antigen was negative. The fecal occult blood test was highly positive, and fecal calprotectin was slightly positive. Upper and lower gastrointestinal (GI) endoscopy showed diffuse atrophic gastric mucosa without the normal folds. Histological examination showed antral gastric mucosa with mild chronic inflammation, whereas the oxyntic mucosa featured severe mucosal atrophy with sclerosis, diffuse pseudo-pyloric metaplasia, and linear hyperplasia of enterochromaffin-like (ECL) cells (Figure 1A). Microbiological investigations for *H pylori* were negative. Moreover, the histological examination showed lamina propria chronic inflammation with eosinophilic infiltration (99 intramucosal eosinophilic granulocytes (eos)/high power field (hpf) in the antrum, 98 eos/hpf in the upper gastric body, and 89 eos/hpf in the lower gastric body) and presence of numerous plasma cells (CD138^+^, MUM1^+^) (Figure 1A(D)). No pathognomonic signs of IBD were observed, but only nodular lymphoid hypertrophy of the last tract of the ileum was found. No parietal cell antibodies (APCA) or anti-IF were found. Vitamin B12 serum levels were normal. Allergy tests were negative. The patient underwent a video capsule endoscopy to complete the diagnostic workup, revealing diffuse epithelial erosion of the duodenum, multiple vascular ectasias, blood vessel fragility, and nodular lymphoid hypertrophy of the terminal ileum. The abdominal ultrasound and entero-MRI findings were negative for bowel thickening and inflammation.

Genetic tests were performed to rule out inherited causes of vascular ectasia and blood vessel fragility. After performing karyotype analysis, which produced normal results, the multi-gene next-generation sequencing revealed that the father inherited a frame-shift mutation in the heterozygous state of the *ABCC6* gene: c.960delC (p.ser321ValfsTer35), compatible with a heterozygous form of pseudoxanthoma *elasticum*.

After the endoscopy, to treat the abnormal eosinophilic inflammation, we decided to start a topical steroid treatment (budesonide 9 mg daily) with progressive decalage in six months. Hereafter, the fecal occult blood test was always negative, and fecal calprotectin normalized, but the previously reported gastric atrophy persisted. The upper and lower GI endoscopies, repeated through the follow-up, confirmed the past histological features with the resolution of the eosinophilic inflammation. IDA was resolved with chronic oral iron supplementation, but iron body stocks were never wholly restored. Therefore, she underwent intravenous supplementation with complete iron status normalization. Considering immunological findings, IgA and IgG serum levels normalized over the years. To date, she is under strict clinical and endoscopic follow-up for a total period of 5 years. The last upper GI endoscopy performed 4 years after the diagnosis revealed the following findings: (i) diffused mild active chronic inflammation and atrophy (severe in the corpus) with sclerosis and antrum gastrin-cell hyperplasia; (ii) corpus lymphoid cells infiltration, pseudo-pyloric metaplasia, and linear ECL-cells hyperplasia; and (iii) fundus simple ECL-cells hyperplasia. Moreover, she is currently taking intramuscular vitamin B12 supplementation.

### 3.2. Case 2

A 12-year-old boy was referred to our Pediatric Clinic for eosinophilic esophagitis (EoE) and suspected AIG (Table 1). He was previously referred to another hospital for nausea, dyspepsia, heartburn, and food impaction episodes not responsive to a proton pump inhibitor (PPI) therapy. His past medical history revealed an IgE-mediated hazelnut allergy. His family history was positive for gastric neoplasia (father and grandfather) and for multiple autoimmune diseases. His father was affected by Basedow–Graves’ disease and presented auto-antibodies against pancreatic and testicular tissues.

The upper GI endoscopy performed at the other center showed a normal esophageal mucosa aspect and hyperemic gastric antral mucosa. The EoE was histologically confirmed (>100 eos/hpf in esophageal biopsies). A gastric mucosa examination revealed mild chronic antral mucosa inflammation, mild atrophy, and linear ECL-cell hyperplasia. A *H pylori* search was negative after Giemsa-stained coloration and immunohistochemical staining with anti-*H. pylori*-specific antibody. He started a treatment with PPI and topical corticosteroid (swallowed fluticasone 875 μg/die). After four months, the upper GI endoscopy was repeated, showing the histological remission of EoE. Gastric biopsies revealed a non-atrophic antral mucosa with gastrin cell hyperplasia, chronic inflammation, and mild atrophy of the fundic mucosa, with focal pseudopyloric metaplasia and linear ECL cell hyperplasia (Figure 1B). The initial diagnostic assessment showed positivity for APCA (1:160). Anemia was not found as an iron status, and vitamin B12 serum levels were normal.

When the child came to our attention, we stopped the PPI treatment and reduced the dose of swallowed steroid (to 500 μg/die) because of EoE remission. Moreover, to exclude the presence of an inborn error of immunity (IEI), we assessed the immunological status, checking immunoglobulins and IgG sub-classes levels that fell within the normal age-matched range.

The esophagogastroduodenoscopy was repeated in our pediatric Center one year after the onset of the symptoms. It confirmed the histological remission of EoE, so we further decreased the amount of swallowed fluticasone (to 250 μg/die). Gastric mucosa was still characterized by moderate atrophy and mild chronic active inflammation of the corpus and the antrum with pseudo-pyloric metaplasia, linear ECL-cell hyperplasia, and corpus lymphoid infiltration.

## 4. Evidence from the Literature Review

We found thirty-nine (39) papers, thirteen of which were selected for the review. Six papers were manually added to the search query due to their relevance, and finally, nineteen papers were analyzed for the review. Pediatric AIG has been diagnosed in 134 children (min 8.5, max 15.7 years).

Among the retrieved articles, pediatric-onset AIG mainly affected females and generally occurred during late childhood and adolescence (Table 2) [2,12,13,14,16,17,18,19,20,21,22,23,24,25,26,27,28,29]. Almost 60% of enrolled patients also presented concomitant autoimmune diseases, particularly AITD and T1DM (20% of cases), followed by CeD (~5%). Addison disease, autoimmune hepatitis, and cytopenia were comorbidities, affecting about 2% of all children [12,13,14,16,17,20,21,23,25,26,27,28,29]. Although family history often did not reveal relevant information, in some cases, there was a positivity for autoimmune diseases [12,20,22]. In the case report by Greenwood et al., family history was positive for autoimmune polyendocrine syndrome type 2 [20].

More than half of the patients presented IDA, which was considered the most common clinical complication of AIG in pediatric patients. Otherwise, megaloblastic anemia was an uncommon presentation (in less than 10% of the patients) [2,12,13,14,16,17,18,19,20,21,22,23,24,25].

APCAs are found in more than two-thirds of the patients [2,12,13,14,16,17,18,20,21,23,24,25,26,27,28,29], while anti-IF antibodies are not always detected (<10% of patients, but not all the patients have been investigated with anti-IF) [2,12,13,20,21,23,25,27,28].

Seronegative AIG (negative APCA screening) was an even rarer clinical entity in children that has been described in eleven patients. Three of these patients had an IEI. One patient had a common variable immunodeficiency (CVID), one CVID plus immune-dysregulation polyendocrinopathy enteropathy X-linked (IPEX) syndrome, and the last was a T-cell primary immunodeficiency [22,23,27]. Moreover, an IEI was identified in four out of twenty-three patients enrolled in a multicenter cohort [28]. Two children had a genetic mutation in the tumor necrosis factor alpha-induced protein 3 (*TNFAIP3*) and cytotoxic T lymphocyte antigen-4 (*CTLA4*) genes. In contrast, the other two had clinical and laboratory features of an IEI without a specific mutation. Interestingly, all of them had concomitant autoimmune comorbidities (i.e., autoimmune enteropathy, autoimmune hepatitis, thyroiditis, Evans’ syndrome, Addison’s disease, CeD, and lichen sclerosis) [16].

Endoscopic features were non-specific (e.g., mucosal erythema, softening of the mucosa, polyps/nodules) and often normal. The histological examination mainly revealed corpus-predominant chronic atrophic gastritis (detected in almost 60% of the patients reported). Still, different reports described more diffuse gastritis/pangastritis (detected in nearly 60% of the patients) [2,12,13,16,17,18,19,20,21,22,23,24]. Pseudopyloric and/or intestinal metaplasia was reported in about half of the patients [2,12,13,16,17,18,19,20,21,22,23,24,27]. The development of hypergastrinemia and achlorhydria was frequently reported during the follow-up, but only a minority (about 40%) of children with AIG presented ECL cell hyperplasia [13,14,16,17,18,19,20,21,23,24]. Moreover, about 25% of patients had a concurrent *H. pylori* infection [2,12,16,20,21,22,23,24]. Gastric adenocarcinomas and type I neuroendocrine tumors were rarely reported [2,12,13,18,19,21,22,24,27,29]. None of the enrolled children was treated with a specific therapy. However, they were often managed using iron and vitamin B12 supplementation [13,16,18,19,20].

## 5. Discussion

We reported two cases of pediatric-onset AIG showing different clinical, serological, and histological presentations.

The first case presented severe and refractory IDA, the most frequent manifestation of AIG in the pediatric age, as confirmed by the literature review [2,12]. IDA is much more common in females, implying that monthly menstrual blood loss may have a further role in its development, aggravated by the inability to increase food-iron absorption due to hypochlorhydria [2]. Our patient tested negative for APCA. Seronegative AIG (APCA negativity) occurs infrequently in pediatric patients with AIG, while it is more common in elderly patients [15,30]. Some patients could be seronegative, especially those with CVID and other IEIs [15,22,23,27,29]. AIG can rarely be associated with selective IgA deficiency [13,14,18]. PA has also been described in patients with other immunodeficiency syndromes, such as chromosome 18q deletion syndrome, X-linked hypogammaglobulinemia, Good syndrome, and CVID [3]. Daniels et al. reported that CVID could affect the GI tract with a broad spectrum of histologic patterns that can mimic lymphocytic gastritis [31]. Nonetheless, AIG could manifest in the context of other IEIs, like *TNFAIP3* and *CTLA4* deficiency, especially with concomitant other autoimmune manifestations [28]. Sometimes, AIG can be incidentally diagnosed during the screening of children with other autoimmune diseases, including T1DM and AITD [12,14,20,21,32]. Especially in patients with T1DM, there is a significant long-term risk of developing AIG [32]. Overall, the most sensitive serum test for AIG is the detection of APCA, but their absence does not exclude a diagnosis of AIG [30]. Even though detecting anti-IF, low pepsinogen, or high gastrinemia levels assists in diagnosing, their sensitivity and specificity are low and need to be integrated with endoscopic and histopathological findings [33].

In case 2, the diagnosis of AIG was incidental and concomitant to the EoE diagnosis. Familiar history is crucial to tailor a patient’s follow-up, particularly regarding neoplastic complications. Patient 2 did not show signs and symptoms of anemia or vitamin B12 deficiency. Pediatric-onset AIG should be considered in the case of refractory IDA and non-specific gastrointestinal disorders non-responsive to conventional treatments. A structured approach is mandatory to ensure comprehensive assessment and management (Figure 2).

A correct serological and endoscopic investigation is needed to differentiate between *H. pylori*-related and autoimmune atrophic gastritis. A comprehensive laboratory assessment is required to evaluate potential AIG-associated conditions and specific warning signs for IEI and APS [2,12,13,14,16,17,18,19,20,21,22,23,24,25,26,27,28]. Finally, a multidisciplinary approach is advised to ensure the optimal management of these patients.

Both our patients showed pathological intestinal eosinophilic inflammation and overlap with eosinophilic gastrointestinal disorder, presenting with non-specific symptoms. Ayaki et al. described a case of a 67-year-old patient with autoimmune polyendocrine syndrome type 2 (APS-2), with both AIG and EoE [34]. No pediatric cases of eosinophilic gastrointestinal diseases and AIG have been reported. The association between AIG and primary eosinophilic gastrointestinal disorders should be further investigated in adult and pediatric populations.

None of our patients had *H. pylori* infection, which should be ruled out when atrophic gastritis is found. Nevertheless, a relatively high prevalence of *H. pylori* is reported in young patients with AIG and microcytic anemia. In the adult model, the causative role of *H. pylori* through a molecular mimicry mechanism still needs to be demonstrated; therefore, no definitive correlations between AIG and *H. pylori* have been found to date [2,17]. Examples of atrophic pangastritis in *H. pylori*-negative patients, frequently associated with other autoimmune disorders, have been recognized. APCA and anti-IF antibodies have been reported in these cases. However, this type of gastritis is usually seen in autoimmune enteropathy and suggests a generalized autoimmune disorder of the gastrointestinal tract or other immunodeficiency disorders (congenital or acquired) [35].

From the genetic point of view, two regions on distal chromosome 4 conferred susceptibility to autoimmune gastritis (*Gasa1* and *Gasa2*). Two minor autoimmune gastritis susceptibility loci have also been identified on chromosome 6 (*Gasa3* and *Gasa4*). AIRE deficiency is also associated with AIG in mouse models and clinical studies on APECED patients (APS1). Of note, the etiology of the AIG that occurs in AIRE-deficient mice is different from that of other mouse models, as the autoantigen targeted is not the H+/K+ ATPase but rather mucin 6 [10]. *CTLA4* and *PTPN22* (APS-2), *ATP4A*, *IL1B*, *IFNGR1*, *LRBA* (IEI due to LRBA deficiency), *RIPK1* (RIPK1-Induced Autoinflammatory syndrome), and several human leukocyte antigens (*HLA*) genes (B-8, B-18, Bw-15, DR-2, DR-4, DR-5, DRB1*03, DRB1*04) increase the risk of developing AIG [10].

The risk of gastric cancer is generally low in pediatric AIG [36]. Several risk factors have been proposed, including the entity of diagnostic delay and disease duration, severity of atrophy and metaplasia, type of metaplasia (complete or incomplete, intestinal or pseudopyloric), presence of epithelial dysplasia, concurrent or previous *H. pylori* infection, concurrent antrum atrophy (atrophic pangastritis), gastric microbiota alterations, and previous gastric surgery [15]. Segni et al. suggested performing upper GI endoscopy with multiple biopsies in patients with AITD, APCA positivity, and hypergastrinemia and endoscopically monitoring patients with AIG every five years, as indicated in the adult population. Moreover, since gastric autoimmunity may occur at any age, they suggested determining APCA every two years in APCA-negative patients with AITD [14]. Despite these findings, recent evidence shows the risk of developing gastric adenocarcinoma without *H. pylori* infection seems not to exceed those of the general population. Any previous or current *H. pylori* infection promotes pangastritis atrophy and is a cancer-promoting cofactor [37]. The same experts also agree that endoscopic surveillance should be recommended at 3–5 years intervals, which is more tailored for the early detection of NETs rather than for gastric cancer secondary prevention. However, only limited data are available [32].

Finally, although increasing knowledge about AIG is available, no specific treatment exists. Therefore, iron and vitamin B12 supplementation are the only therapeutic strategies to avoid nutritional and metabolic complications [15].

## 6. Conclusions

Pediatric onset AIG is a rare disease with a proteiform spectrum of clinical manifestations that range from incidental endoscopic findings to severe IDA. Diagnosis can be challenging and often delayed. Many comorbidities can be associated with pediatric AIG, including autoimmune diseases, IEI, or rarely, eosinophilic gastrointestinal disorders. The diagnostic workup of pediatric AIG needs a multistep approach and a multidisciplinary team that involves a gastroenterologist, endoscopist, pathologist, nutritionist, immunologist, and family pediatrician. A periodic follow-up and the early identification of potential complications (gastric cancer, autoimmune diseases, and immunodeficiencies) are pivotal for managing these children accurately.

## Figures and Tables

**Figure 1 diseases-13-00133-f001:**
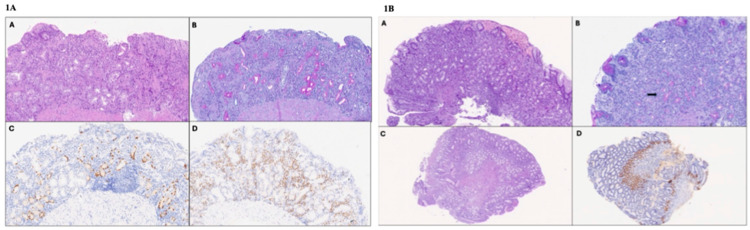
(**1A**) **Histologic images of corpus biopsies of Case 1 (high-power field 40×).** (**A**) The oxyntic mucosa shows moderate chronic inflammation, including eosinophils and severe atrophy, with a reduction in the number of appropriate glandular units, fibrosclerosis, and metaplastic changes in the epithelium (hematoxylin and eosin). (**B**) Diffuse pseudopyloric (neutral mucin-producing) metaplasia of the oxyntic glands (Periodic Acid Schiff–Alcian Blue staining). (**C**) Linear ECL cell hyperplasia (chromogranin A immunostaining). (**D**) Presence of numerous MUM1-positive plasma cells in the lamina propria (MUM1 immunostaining). (**1B**) **Histologic images of gastric biopsies of Case 2 (high-power field 40×)**. (**A**) The fundic mucosa features moderate chronic inflammation and mild atrophy (hematoxylin and eosin). (**B**) Focal pseudopyloric metaplasia (arrow; Periodic Acid Schiff–Alcian Blue staining). The antral mucosa appears normotrophic (**C**) (hematoxylin and eosin), with gastrin cell hyperplasia (**D**) (gastrin immunohistochemistry).

**Figure 2 diseases-13-00133-f002:**
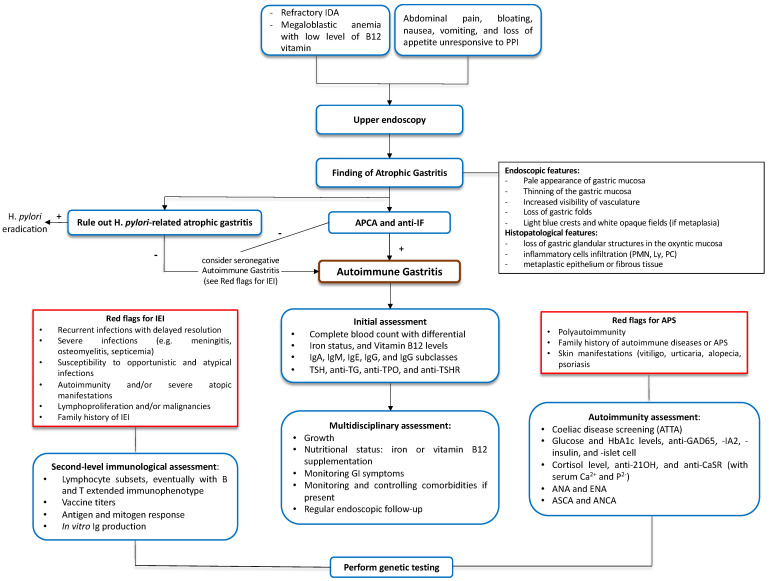
**Suggested diagnostic flowchart for pediatric autoimmune gastritis.** ANA, anti-nuclear antibodies. ANCA, anti-neutrophil cytoplasmic antibodies. Anti-21OH, anti-21-hydroxylase antibodies. Anti-CaSR, anti-Calcium Sensing Receptor antibodies. Anti-GAD65, anti-glutamic acid decarboxylase 65-kilodalton isoform antibodies. Anti-IA2, anti-tyrosine phosphatase-related islet antigen two antibodies. Anti-IF, anti-intrinsic factor antibodies. Anti-TG, anti-thyroglobulin antibodies. Anti-TPO, anti-thyreoperoxidase antibodies. anti-TSHR, anti-thyroid-stimulating hormone receptor. APCA, anti-parietal cell antibodies. APS, autoimmune polyglandular syndrome. ASCA, anti-*Saccharomyces cerevisiae* antibodies. ATTA, Anti-Tissue Transglutaminase Antibodies. ENA, anti-extractable nuclear antigen antibodies. *H. pylori*, *Helycobacter pylori*. HbA1c, hemoglobin A1c. IDA, iron-deficiency anemia. IEI, inborn error of immunity. Ig, immunoglobulin. Ly, lymphocyte. PC, plasma cells. PMN, neutrophils. PPI, proton pump inhibitor. TSH, thyroid-stimulating hormone.

**Table 1 diseases-13-00133-t001:** Main clinical and laboratory findings of the cases presented.

CASE 1	Main Findings
**Symptoms**	Asthenia and weight loss. Severe IDA.
**Laboratory findings**	Hb 7 g/dL ^†^, MCV 59.1 fl ^†^, transferrin saturation 1% ^†^, ferritin 1 ng/mL ^†^. Vit. B12 serum levels: 266 pg/mL; folate serum levels: 8.10 ng/mL. IgA 56 mg/dL ^†^, IgG 348 mg/dL ^†^, IgM 66 mg/dL, IgG1 287.3 mg/dL ^†^, IgG2 83 mg/dL ^†^, IgG3 21.5 mg/dL, IgG4 50.2 mg/dL. CD3+ cells 1751/mm^3^, CD4+ 1.009/mm^3^, CD8+ 536/mm^3^, CD19+ 185/mm^3 †^. Anti-HBsAg: 24.4 mIU/mL (presence of immunity); IgG anti-measles: 144 AU/mL presence of immunity); IgG anti-tetanus: 0.440 IU (vaccine protection); IgG anti-diphtheria: 0.142 IU (vaccine protection). ANA: < 1:80; ENA test: negative; ASCA: IgA 0.4 IU/mL, IgG 0.6 IU/mL. ANCA: IIF < 1:20; anti-MPO: 0.2 IU/mL, anti-PR3: 0.2 IU/mL. anti-TPO: 13.1 mIU/mL, anti-TG: <20 mIU/, ATTA-reflex test: negative. APCA: < 1:40. Anti-IF: not present. Anti-insulin ab: 1.6 IU/mL; anti-pancreatic islet ab: <1:4 IU/mL. *H. pylori* fecal antigen: negative. Fecal occult blood test: positive. Fecal calprotectin: 144.7 mg/kg ^†^ (0–50 mg/kg). Stool culture for BK: negative. IGRA: negative. Ab anti-*Strongyloides* (ELISA): negative. Ab anti-*Toxocara* (ELISA and Immunoblotting): negative. Ab anti-*Trichinella* (immunoblotting): negative. Ova and parasite stool exam: negative. Graham test: negative.
**Endoscopy and histological findings**	Diffuse atrophic gastric mucosa without the normal folds. Antral gastric mucosa with mild chronic inflammation, and severe atrophy of the oxyntic mucosa with sclerosis, pseudo-pyloric metaplasia, and simple and linear hyperplasia of ECL cells. Nodular lymphoid hypertrophy of the last tract of the ileum. VCE: diffuse inflammation of the small intestine with epithelial erosion, multiple vascular ectasias, and blood vessel fragility.
**Treatment**	Iron and vitamin B12 supplementation.
**CASE 2**	**Main findings**
**Symptoms**	Incidental finding during upper GI endoscopy for PPI non-responsive nausea, dyspepsia, heartburn, and severe food impaction episodes.
**Laboratory findings**	Hb 12.5 g/dL, MCV 82.9 fl, transferrin saturation 14%, ferritin 18.8 ng/mL. Vitamin B12 serum levels: 381 pg/mL; folate serum levels: 6.20 ng/mL. IgA 226 mg/dL, IgG 1115 mg/dL, IgM 136 mg/dL, IgG1 566.3 mg/dL, IgG2 285.3 mg/dL, IgG3 96.9 mg/dL, IgG4 19.9 mg/dL. APCA: 1:320 ^†^. Anti-IF: not present. ANA: <1:80; ENA screening test: negative; anti-TPO: 13.1 mIU/, anti-TG: 20 mIU/mL, ATTA-reflex test: negative. Gastrin: 15.2 pg/mL [<108 pg/mL]. Chromogranin A: 100 ng/mL [19.4–98 ng/mL]. *H. pylori* fecal antigen: negative.
**Endoscopy and histological findings**	Regular esophageal mucosa and hyperemic gastric antral mucosa. Esophagus: intraepithelial eosinophilic granulocytes (>100 eos/hpf). Giemsa-stained special coloration for *H. pylori* and immunohistochemical staining with anti-*H. pylori* search negative.
**Treatment**	Topical corticosteroid (swallowed fluticasone; starting dose 875 μg/die with progressive *decalage*)

^†^ = abnormal value. Ab, antibodies. ANA, antinuclear antibodies. ANCA, anti-neutrophil cytoplasmic antibodies. Anti-HBsAg, anti-Hepatitis B surface antigen antibodies. Anti-IF, autoantibodies against Intrinsic Factor. Anti-MPO, anti-myeloperoxidase antibodies. APCA, anti-parietal cells antibodies. anti-PR3, anti-proteinase3 antibodies. anti-TG, anti-thyroglobulin antibodies. Anti-TPO, anti-thyroid peroxidase antibodies. ASCA, anti-*Saccharomyces cerevisiae* antibodies. ATTA, Anti-Tissue Transglutaminase Antibodies. BK, *Mycobacterium tuberculosis*. ECL, enterochromaffin-like. ELISA, enzyme-linked immunosorbent assay. ENA, extractable nuclear antigen test. GI, gastrointestinal. *H. pylori*, *Helicobacter pylori*. *Hb*, hemoglobin. IDA, iron deficiency anemia. Ig, immunoglobulin. IGRA, Interferon Gamma Release Assay. IIF, indirect immunofluorescence. MCV means cellular volume. PPI, proton pump inhibitor. VCE, video capsule endoscopy.

**Table 2 diseases-13-00133-t002:** Summary of the pediatric autoimmune gastritis reports.

Author, Year	Type of Study	N°, Sex	Mean Age at the Diagnosis	Comorbidities	Iron Deficiency	B12 Deficiency	Hyper-Gastrinemia	APCA	Anti-IF	Macroscopic and Histologic Findings	HP Infection	Metaplasia	ECL Cell Hyperplasia	Therapy
Katz et al., 1997 [19]	Case report	M	15	No	Yes	Yes	/	/	/	Atrophic gastric mucosa with antrum polyp. CSP	/	Yes (intestinal). Adenoma and poorly differentiated adenocarcinoma.	No	Oral iron supplementation + IM vitamin B12 Surgery.
Segni et al., 2004 [14]	Case series	7 F (2) M (5)	13	CLT (5/7), GD (2/7), T1DM (1/7)	/	/	Yes (6/7)	+ (7/7)	/	CSP (3/7), CPAG (4/7)	Yes (4/7)	/	/	/
Greenwood et al., 2008 [20]	Case series	2 M (1) F (1)	8.5	AD (1/2), CLT (2/2), and alopecia (1/2).	Yes (2/2)	Yes (1/2)	Yes (1/2)	+ (2/2)	+ (1/2)	Mucosal erythema (1/2). Gland distortion and nodular lymphoid aggregate (1/2). Chronic inflammatory cell infiltration (lymphocytes with occasional PC and Eos) (1/2).	No (2/2)	Yes (intestinal) (1/2)	No	Oral iron and vitamin B12 supplementation (2/2)
Frohlich-Reiterer et al., 2011 [21]	Cross-sectional	3 F (3/3)	15.7	T1DM (3/3), CLT (3/3)	Yes (3/3)	No	Yes (2/3)	+ (3/3)	/ (3/3)	Mild atrophic gastritis (1/3); mild CSP (2/3).	Yes (2/3)	No (3/3)	No (3/3)	/
Russell et al., 2012 [22]	Case report	F	15	No	No	No	/	-	-	Macroscopic nodules. Multifocal atrophic gastritis.	No	Yes (focal, intestinal)	Yes	/
Gonçalves et al., 2014 [12]	Case series	5 M (2) F (3)	13.6	T1DM (1/5), CLT (1/5)	Yes (5/5)	No (3/5); / (2/5)	Yes (3/5)	+ (5/5)	/ (5/5)	Fold softening (2/5). CPAG [5/5; mild (1/5), moderate to severe (4/5)].	No	Yes [4/5; intestinal (1/5); pseudopyloric (3/5)]	Yes – linear (4/5) – nodular (3/5)	/
Pogoriler et al., 2015 [23]	Case series	12 M (6) F (6)	11.6	T1DM (4/12), AI hepatitis (1/12), AI cytopenia (2/12), CeD (1/12), CLT (2/12), CVID (1/12), T-cell PID (1/12)	Yes (6/12)	Yes (1/9); / (3/12)	Yes (2/2); / (10/12)	+ (3/7) *; / (5/12).	+ (2/5); / (7/12)	Not specific (12/12). Chronic inflammation: – antrum (9/12); – oxyntic mucosa (12/12).	No	Yes [8/12; intestinal (3/8); pseudopyloric (4/8); squamous-mucinous (1/8)]. One patient developed gastric adenocarcinoma.	Yes (5/12)	/
Kirsaclioglu et al., 2014 [18]	Case report	F	14	No	Yes	/	Yes	+	/	Corpus polyp. Chronic atrophic gastritis.	/	Yes (intestinal)	Type 1 GCT	EMR
Miguel et al., 2014 [16]	Case series	8 M (2) F (6)	12.3	ANA positivity (1/8)	Yes (8/8)	No (5/5); / (3/8)	Yes (8/8)	+ (8/8)	/ (8/8)	Chronic atrophic gastritis: – antrum (5/8) – antrum and corpus (2/8) – antrum, corpus, and fundus (1/8)	Yes (4/8)	Yes [1/8 (intestinal)]	/ (8/8)	Oral iron supplementation (8/8)
Koca et al., 2016 [24]	Case report	F	15	No	/	/	Yes	+	/	Nodules. Multifocal atrophic gastritis	Yes	Yes (intestinal)	Yes	/
Besançon et al., 2017 [25]	Case series	2 M (1), F (1)	12	T1DM (2/2), CLT (1/2),	Yes (1/2)	/	/	+ (2/2)	/	Chronic lymphocytic gastritis of the fundus (1/2). Gastritis of the antrum and fundus (1/2).	Yes (1/2)	/	/	Oral iron supplementation
Saglietti et al., 2018 [2]	Case series	2 F (2/2)	14.5	No	Yes (2/2)	Yes (1/2); / (1/2)	/ (2/2)	+ (2/2)	/ (2/2)	Normal aspect (2/2). Multifocal atrophic gastritis (2/2).	No	Yes (2/2) Intestinal (1/2) Pseudopyloric (1/2)	Yes (2/2)	/
Moreira-Silva et al., 2019 [17]	Case series	20 M (9) F (11)	12.3	CLT (5/20), T1DM (4/20), CeD (1/20), ITP (1/20), FSGS (1/20), IgAGN (1/20)	Yes (18/20)	/ (20/20)	Yes (19/20)	+ (20/20)	/ (20/20)	Chronic inflammation of the gastric mucosa in the corpus	Yes (11/20)	Yes (4/20, intestinal)	/	/
Calcaterra et al., 2020 [26]	Single-center retrospective study	1 (F)	16	CLT	Yes	No	No	+	/	Foveolar hyperplasia	No	/	/	/
Mitsinikos et al., 2020 [13]	Case series	3 F (3/3)	14	T1DM (2/3), AD (1/3), CLT (1/3), crescentic GN (1/3).	Yes (3/3)	Yes (1 */3) * pancytopenia	Yes (1/3)	+ (3/3)	+ (1/3)	Normal aspect (1/3), linear furrowing (1/3), erythema (1/3). Oxyntic mucosa mononuclear cells infiltration and gland damage, and decreased parietal cell mass (3/3)	No	Yes [2/3: pseudopyloric (1/2), intestinal (1/2)].	Yes [2/3: Linear (2/3)].	Oral iron (2/3) and vitamin B12 supplementation (1/3).
Demir et al., 2020 [29]	Single-center retrospective-observational study	10 F (4) M (6)	15	CLT (1/10), CeD (1/10)	Yes (5/10)	/	/	+ (10/10)	/	Hyperemic gastric corpus (5/10) or antrum (5/10). Gastric inflammation (7/10)	No	Yes (4/10, intestinal)	Yes (1/10)	/
Kulak et al., 2021 [27]	Single-center retrospective study	22 F (15) M (7)	10.9	CLT (3/22), GD (2/22), T1DM (3/22), CVID and IPEX (1/22), IgA deficiency (1/22), AI hepatitis (1/22), CeD (1/22), Crohn’s disease (1/22), JIA (1/22)	Yes (4/22)	Yes (1/22)	Yes (8/10 tested)	+ (6/12 tested)	+ (0/7 tested)	CPAG (22/22), antral atrophy (4/22)	Yes (3/5 tested)	Yes [4/22: pseudopyloric (3/4), intestinal (1/4)]	Yes (22/22)	/
Granot et al., 2024 [28]	Multicenter retrospective study	33 F (23) M (10)	12 [median; IQR 7.0–15.1]	CeD (3/33), AITD (6/33), T1DM (4/33), AI hepatitis (2/33), AD (1/33)	Yes (25/33)	Yes (2/33)	Yes (23/27 tested)	+ (16/33)	+ (6/33)	Active inflammation: gastric body (5/28 tested), gastric antrum (19/31); Chronic inflammation: gastric body (18/30 tested), gastric antrum (27/28);	Yes (1/33)	Yes (8/32 gastric antrum: Pseudo pyloric 5/8; intestinal 1/8; both 2/8)	Yes (20/34)	Iron supplementation: oral (17/33) and IV (6/33); vit. B12 oral supplementation (5/33)

/, not available. +, presents; AD, Addison disease. AI, autoimmune. AITD, autoimmune thyroiditis. ANA, antinuclear antibody. APCA, anti-parietal cells antibodies. CeD, coeliac disease. CLT, chronic lymphocytic thyroiditis. CPAG, Corpus predominant atrophic gastritis. CSP, Chronic superficial pangastritis. CVID, Common Variable Immuno-Deficiency. ECL, enterochromaffin-like. Eos, eosinophils. EMR, endoscopic mucosal resection. F, female. FSGS, focal segmental glomerular sclerosis. GCT, gastric carcinoid tumors. GD, Graves’ disease. GN, glomerulonephritis. HP, *Helicobacter pylori*. IM, intra-muscular. IQR, inter-quartile range. IPEX, Immune-dysregulation Poliendocrinopahty Enteropathy X-linked. ITP, immune thrombocytopenia. JIA, juvenile idiopathic arthritis. M, male. NOI, nothing of interest. PC, plasma cells. PID, primary immunodeficiency. T1DM, type 1 diabetes mellitus. * *Note*: of the four patients with APCA –, one was diagnosed with CVID and one with T-cell PID.

## Data Availability

The dataset generated and analyzed during the current study is available from the corresponding author upon reasonable request (ivan.taietti@gmail.com).

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
