# Peer review of "Clinical Heterogeneity of Early-Onset Autoimmune Gastritis: From the Evidence to a Pediatric Tailored Algorithm"

_diseases, 2025, doi:10.3390/diseases13050133_

Round 1
Reviewer 1 Report
Comments and Suggestions for Authors
Dear Authors,
Thank you for submitting the manuscript entitled "Case report Clinical heterogeneity of early-onset autoimmune gastritis: from the evidence to a pediatric tailored algorithm”. The manuscript covers very interesting and important topic. However, there are some open questions that need to be clarified:
- Introduction
- What was the aim of this case report?
- Lines 64-72:
- Was it a prospective study or was it decided to present the patients after the diagnosis was established? Were data at diagnosis collected retrospectively?
- What did parents sign the informed consent to?
- The last sentence (lines 71-72): „The Ethical Committee of Fondazione IRCCS San Matteo di Pavia, Italy.“ seems incomplete
- Please provide more data on literature review – how many papers were found, were some excluded and why; consider showing the flowchart
- Detailed Case Description
- Case 1.
- Consider using different abbreviation for celiac disease as B-lymfocytes are also CD
- Line 100 what does „positiveecal“ mean?
- Please explain what is meant by „Abdominal US was negative.“ – for what?
- For how long was the patient followed-up? What was the result of the most recent endoscopy and when it was performed?
- Case 2.
- „He was born after a pregnancy medically induced by in vitro fertilization.“ Is not relevant for the manuscript
- pylori abbreviation should be explained
- Were the gastric biopsies taken at the first GI endoscopy performed at another center and if yes what was the result?
- How did gastric mucosa look at the repeated esophagogastroduodenoscopy one year after the symptom's onset and what did histology show?
Table 1
- Please explain ATTA-reflex test
- Please explain the abbreviation H. pylori (or use the whole word as previously in the text – please unify)
- Pylori should be written in italics and Pylori as pylori
- Some abbreviations are redundant (EREFS, PEESS) and some are not explained (e.g, Ig, anti-HBs, ECL)
- Culture is misspelled (colture)
- In the text it is said that patient 2 was also taking IPP
- PCA or APCA?
- Consider either showing normal values for laboratory results or just stating was it normal or abnormal
Table 2
- please describe abbreviations M, F, IM, ECL, Ig, IPEX, IQR, AITD
- what is meant by Chron disease?
Figure 2
- what is meant by complete blood count with formula?
- Please describe abbreviations IDA, PPI, IEI, APS
- pylori should be written in italics
- Is it possible to make a diagnosis of AIG even if APCA and anti-IF are negative?
- Discussion
3.1. Evidence from the literature
- In general – too many repetition of data from the table, and insufficient discussion
- Does the statement that “AIG mainly affected females and generally occurred during late childhood and adolescence” refers only to pediatric population or to general population?
- How many patients in total have been described, what was the age range?
- “Three patients had an IEI” – out of how many?
- “Moreover, IEI were identified in four patients among the multicenter cohort…” – again out of how many?
- Consider being more specific than using mostly, often, frequently, only a minority, several, rarely…
3.2. Discussion
- Again, too many repetition of patients’ data
- Consider commenting how the diagnosis of AIG was made if APCA was negative
- Consider discussing treatment options
- Presented were two patients at the time of diagnosis, not much was said about follow-up and the algorithm and conclusions are mainly on follow-up – please comment
- Conclusions
- the last sentence is not clear
References
- please use the consistent formatting of the references
- name of the journal shoud be abbreviated – please correct ref. 2, 4-10, 12, 13, 22
- English language needs editing
Author Response
Point by point response
REVIEWER 1
“Dear Authors,
Thank you for submitting the manuscript entitled "Case report Clinical heterogeneity of early-onset autoimmune gastritis: from the evidence to a pediatric tailored algorithm”. The manuscript covers very interesting and important topic. However, there are some open questions that need to be clarified”
The authors thank the reviewer for the comments and suggestions that improved the manuscript quality.
- Introduction
“What was the aim of this case report?”
We thank the reviewer for the comment. We added a sentence about the specific aim of our case description (Lines 63-64).
“Lines 64-72”: “Was it a prospective study or was it decided to present the patients after the diagnosis was established? Were data at diagnosis collected retrospectively?”
We thank the reviewer for the comment. We decided to present the patient’s data after we established the diagnosis. The data have been collected retrospectively. Please see line 83.
“What did parents sign the informed consent to?”
We thank the reviewer for the comment. The parents signed the informed consent form for scientific research and publication (Lines 90-91).
“The last sentence (lines 71-72): „The Ethical Committee of Fondazione IRCCS San Matteo di Pavia, Italy.“ seems incomplete.”
We apologize for the mistake. We corrected the text; please see lines 90-91.
“Please provide more data on literature review – how many papers were found, were some excluded and why; consider showing the flowchart”
We thank the reviewer for the comment. We would clarify that we did not perform a systematic review, but we clarified the literature search strategy in the method and results sections.
- Detailed Case Description
“Case 1.”
“Consider using different abbreviations for celiac disease as B-lymfocytes are also CD”
We thank the reviewer for the comment. We used a new abbreviation in the track changes, namely “CeD”
“Line 100 what does „positiveecal“ mean?”
We thank the reviewer for the comment. We corrected the text.
“Please explain what is meant by „Abdominal US was negative.“ – for what?”
We thank the reviewer for the comment. We clarified this point in the text; please, see case 1.
“For how long was the patient followed-up? What was the result of the most recent endoscopy and when it was performed?”
We thank the reviewer for the comment. We added the required details in Lines 142-147.
Case 2.
“He was born after a pregnancy medically induced by in vitro fertilization.“ Is not relevant for the manuscript”
We thank the reviewer for the comment.
“pylori abbreviation should be explained”
We thank the reviewer for the comment. All the explanations are in the track changes.
“Were the gastric biopsies taken at the first GI endoscopy performed at another center and if yes what was the result?”
We thank the reviewer for the comment. We clarified this point; please see Case 2.
“How did gastric mucosa look at the repeated esophagogastroduodenoscopy one year after the symptom's onset and what did histology show?”
We thank the reviewer for the comment. We clarified this point; please see Case 2.
Table 1
“Please explain ATTA-reflex test”
We added this explanation to the table 1.
“Please explain the abbreviation H. pylori (or use the whole word as previously in the text – please unify)”
We added this explanation [i.e. Helicobacter pylori (H. pylori)] to the table 1.
“Pylori should be written in italics and Pylori as pylori”
We provided the correction suggested.
“Some abbreviations are redundant (EREFS, PEESS) and some are not explained (e.g, Ig, anti-HBs, ECL)”
We modified the abbreviation according to the reviewer’s suggestions.
“Culture is misspelled (colture)”
We corrected the spelling.
“In the text it is said that patient 2 was also taking IPP”
We clarified this point in table 1.
“PCA or APCA?”
We thank the reviewer for the comment and provide all the needed corrections in the text.
“Consider either showing normal values for laboratory results or just stating was it normal or abnormal”
We thank the reviewer for the comment. We added “†” at the apex of the values that resulted as abnormal, specifying its meaning in the Table caption.
Table 2
“please describe abbreviations M, F, IM, ECL, Ig, IPEX, IQR, AITD”
We added the explanations according to the reviewer’s suggestions.
“what is meant by Chron disease?”
We corrected the spelling in the table.
Figure 2
“what is meant by complete blood count with formula?”
We corrected Figure 2, adding “complete blood count with differential”.
“Please describe abbreviations IDA, PPI, IEI, APS”
We modified this point in the text. All the explanations suggested are reported in Figure 2’s caption.
“pylori should be written in italics”
We corrected the figure.
“Is it possible to make a diagnosis of AIG even if APCA and anti-IF are negative?”
We thank the reviewer for the comment. We explain this point in the “Discussion” paragraph.
Discussion
3.1. Evidence from the literature
“In general – too many repetition of data from the table, and insufficient discussion”
We thank the reviewer for the comment. Our goal was to summarize the available evidence in the text while providing more specific details in the table to highlight the characteristics of the different reports. We kept the discussion brief in this paragraph, as we planned to explore it in greater detail in Discussion.
“Does the statement that “AIG mainly affected females and generally occurred during late childhood and adolescence” refers only to pediatric population or to general population?”
We clarified this point in the text; please see the paragraph 4.
“How many patients in total have been described, what was the age range?”
To date, 134 pediatric patients have been reported. We did not perform a systemic review and meta-analysis. Due to the heterogeneity of the reports, we were not able to calculate the specific age range. Please see lines 198-199.
“Three patients had an IEI” – out of how many?”
The three patients with IEI were identified among the eleven with seronegative AIG, as reported in the literature review. Please see lines 209-213.
“Moreover, IEI were identified in four patients among the multicenter cohort…” – again out of how many?”
These patients were identified out of twenty-three patients. Please see lines 213-2016.
“Consider being more specific than using mostly, often, frequently, only a minority, several, rarely…”
We thank the reviewer for the comment. Since the reports are heterogeneous and no systematic review with meta-analysis has been conducted, we cannot define the exact percentages. Table 1 aims to show the specific relative percentages of the different literature reports.
3.2. Discussion
Again, too many repetitions of patients’ data
We revised the text according to the reviewer’s suggestion.
“Consider commenting how the diagnosis of AIG was made if APCA was negative”
We thank the reviewer for the comment. We provide the requested explanation. Please see Discussion section.
“Consider discussing treatment options”
We briefly discussed on it in the paragraph.
“Presented were two patients at the time of diagnosis, not much was said about follow-up and the algorithm and conclusions are mainly on follow-up – please comment”
We modified the text according to the reviewer’s suggestions. Please see the case presentation and discussion sections.
Conclusions
the last sentence is not clear
We improved the conclusion section.
References
- “please use the consistent formatting of the references”
- “Name of the journal shoud be abbreviated – please correct ref. 2, 4-10, 12, 13, 22”
We thank the reviewer for the comment. We changed the format of the references.
Reviewer 2 Report
Comments and Suggestions for Authors
In recent decades, there has been a tendency for the proportion of Helicobacter pylori infection to decrease in the structure of etiologic factors of chronic gastritis, which is due, on the one hand, to the success of eradication therapy, and on the other hand, to a decrease in the number of new cases of infection. At the same time, the importance of other causes of the disease, which were previously considered extremely rare, is increasing. Currently, international consensus documents draw the attention of clinicians to the increasing prevalence of autoimmune gastritis (AIG) in the population. At the same time, in real clinical practice, autoimmune gastritis is often diagnosed already at the stage of severe atrophy and formed deficiency of micro- and macronutrients, requiring correction of the nutritional status to prevent the development of significant neurological disorders, anemic syndrome, and osteopenia/osteoporosis. Therefore, increasing knowledge among clinicians about AIG, its clinical manifestations, diagnostic methods and possible treatment approaches is extremely important. This emphasizes the importance of this publication. At the same time, I would like to give several wishes to the authors. In the introduction, you write that AIH increases the risk of developing neuroendocrine tumors and gastric adenocarcinoma. However, currently published consensuses, in particular REGAIN, indicate that the risk of developing gastric adenocarcinoma in the absence of current or past Helicobacter pylori infection does not exceed the general population risk. I would like to see data included in accordance with this concept. There are also a number of questions regarding the described clinical cases. Were the patients tested for gastropanel, antibodies to Castle factor? Was immunohistochemical examination of gastrobiopsies performed to assess enterochromaffin cell hyperplasia? In one of the cases, involvement of the antral region in the inflammatory process was described; is it necessary to clarify the etiologic factor? Also, in one of the clinical cases, budesonide was prescribed for 6 months; what was the effect of this therapy?
Author Response
Point by point response
REVIEWER 2
“In recent decades, there has been a tendency for the proportion of Helicobacter pylori infection to decrease in the structure of etiologic factors of chronic gastritis, which is due, on the one hand, to the success of eradication therapy, and on the other hand, to a decrease in the number of new cases of infection. At the same time, the importance of other causes of the disease, which were previously considered extremely rare, is increasing. Currently, international consensus documents draw the attention of clinicians to the increasing prevalence of autoimmune gastritis (AIG) in the population. At the same time, in real clinical practice, autoimmune gastritis is often diagnosed already at the stage of severe atrophy and formed deficiency of micro- and macronutrients, requiring correction of the nutritional status to prevent the development of significant neurological disorders, anemic syndrome, and osteopenia/osteoporosis. Therefore, increasing knowledge among clinicians about AIG, its clinical manifestations, diagnostic methods and possible treatment approaches is extremely important. This emphasizes the importance of this publication. At the same time, I would like to give several wishes to the authors.”
“In the introduction, you write that AIH increases the risk of developing neuroendocrine tumors and gastric adenocarcinoma. However, currently published consensuses, in particular REGAIN, indicate that the risk of developing gastric adenocarcinoma in the absence of current or past Helicobacter pylori infection does not exceed the general population risk. I would like to see data included in accordance with this concept.”
We thank the reviewer for the comment. We explain this concept in the Discussion.
“There are also a number of questions regarding the described clinical cases”.
- “Were the patients tested for gastropanel, antibodies to Castle factor?”
We thank the reviewer for the comment. The patients were tested for an autoimmunity “gastropanel” comprehensive of APCA and anti-IF (or Castle factor) antibodies. Please see Table 1 and the case presentation.
- “Was immunohistochemical examination of gastrobiopsies performed to assess enterochromaffin cell hyperplasia?”
We thank the reviewer for the comment. Yes, an immunohistochemical examination of gastrobiopsies was performed to assess ECL hyperplasia. Please see Table 1 and the case presentation.
- “In one of the cases, involvement of the antral region in the inflammatory process was described; is it necessary to clarify the etiologic factor?”
We thank the reviewer for the comment. In the case of patient 1, where we found inflammation of the antral region, we ruled out H. pylori etiology. Please see Case 1.
- “Also, in one of the clinical cases, budesonide was prescribed for 6 months; what was the effect of this therapy?”
We thank the reviewer for the comment. Patient 1 was treated with oral budesonide for 6 months (6 mg/daily with progressive tapering), whose indication was the abnormal eosinophilic inflammation of the gastric mucosa. Budesonide resolved eosinophilic inflammation and the previous finding of positive occult stool blood. Please, see case 1.
Round 2
Reviewer 1 Report
Comments and Suggestions for Authors
Dear Authors,
Thank you for submitting the revised manuscript entitled "Clinical heterogeneity of early-onset autoimmune gastritis: from the evidence to a pediatric tailored algorithm” and for answering the questions. However, there are still some open questions that need to be clarified:
Table 1
- Pylori should be written in italics and Pylori as pylori
Figure 2
- Please describe abbreviations APCA, anti-IF, APS, TSH, anti-TG, anti-TPO, anti-TSHR, ANA, ENA, ASCA, ANCA
- Is it possible to make a diagnosis of AIG even if APCA and anti-IF are negative?
- according to the figure, if APCA and anti-IF are negative AIG is excluded, and one of the presented AIG patients was negative for APCA and anti-IF – consider showing this possibility in the figure
- should pylori be excluded before APCA and anti-IF are performed (in the text it is said: “None of our patients had H. pylori infection, which should be ruled out when atrophic gastritis is found.”
- Evidence from the literature
- regarding using terms such as mostly, often, frequently, only a minority, several, rarely… thank you for your answer. However, although the reports are heterogeneous and no systematic review with meta-analysis has been conducted, you somehow reached the terms of often, frequently and so on – please explain based on what and where in Table 1 are percentages
4.. Discussion
- What is meant by N?
Author Response
Point by point response – Round 2
REVIEWER 1
“Dear Authors,
Thank you for submitting the revised manuscript entitled "Clinical heterogeneity of early-onset autoimmune gastritis: from the evidence to a pediatric tailored algorithm” and for answering the questions. However, there are still some open questions that need to be clarified”
The authors thank the reviewer for the comments and suggestions that improved the manuscript quality.
Table 1
- “Pylori should be written in italics and Pylori as pylori”
We thank the reviewer for the comment. We correct the text in the caption of “Table 1” (see line 221)
Figure 2
- “Please describe abbreviations APCA, anti-IF, APS, TSH, anti-TG, anti-TPO, anti-TSHR, ANA, ENA, ASCA, ANCA”
We thank the reviewer for the comment. We correct the text in the caption of “Figure 2” (see lines 310-320)
- “Is it possible to make a diagnosis of AIG even if APCA and anti-IF are negative? According to the figure, if APCA and anti-IF are negative AIG is excluded, and one of the presented AIG patients was negative for APCA and anti-IF – consider showing this possibility in the figure.”
We thank the reviewer for the comment. We improve the figure with the given suggestion (see the new “Figure 2” page 11)
- “Should pylori be excluded before APCA and anti-IF are performed (in the text it is said: “None of our patients had H. pylori infection, which should be ruled out when atrophic gastritis is found.”
We thank the reviewer for the comment. We suggest to simultaneously rule out H. pylori infection and check APCA and anti-IF when gastric atrophy is found.
Evidence from the literature
- “Regarding using terms such as mostly, often, frequently, only a minority, several, rarely… thank you for your answer. However, although the reports are heterogeneous and no systematic review with meta-analysis has been conducted, you somehow reached the terms of often, frequently and so on – please explain based on what and where in Table 1 are percentages”
We thank the reviewer for the comment. For each feature and clinical/bio-humoral point discussed in the text, we referred to the specific paper (case reports/series, cohort studies) that has also been referred to in “Table 1”.
Discussion
- “What is meant by N?”
We thank the reviewer for the comment. We correct the text (see line 286).
Round 3
Reviewer 1 Report
Comments and Suggestions for Authors
Dear Authors,
Thank you for submitting the revised manuscript entitled "Clinical heterogeneity of early-onset autoimmune gastritis: from the evidence to a pediatric tailored algorithm” and thank you for improving the manuscript and answering to the comments. However, please consider also the following:
Figure 2
- Regarding the comment to consider showing this possibility of APCA and anti-IF negative case– thank you for adding this to the figure. However, the line between APCA and anti-IF is redundant – there is no need to have two lines for pylori while the comment “consider seronegative autoimmune gastritis” deserves separate box and the line from APCA and anti-IF box
Evidence from the literature
- regarding using terms such as mostly, often, frequently, only a minority, several, rarely… thank you for your answer. However, in my opinion, the way it is written it does not give strength to what is presented, it is far from precise and not easy to read
Author Response
Point by point response – Round 3
REVIEWER 1
“Dear Authors,
Thank you for submitting the revised manuscript entitled "Clinical heterogeneity of early-onset autoimmune gastritis:
from the evidence to a pediatric tailored algorithm” and thank you for improving the manuscript and answering to the
comments. However, please consider also the following:”
The authors thank the reviewer for the comments and suggestions that have greatly improved the manuscript quality.
§ Figure 2
- “Regarding the comment to consider showing this possibility of APCA and anti-IF negative case– thank you
for adding this to the figure. However, the line between APCA and anti-IF is redundant – there is no need to
have two lines for pylori while the comment “consider seronegative autoimmune gastritis” deserves separate
box and the line from APCA and anti-IF box”
We thank the reviewer for the comment. We improve the figure with the given suggestion (see Figure 2)
§ Evidence from the literature
- “Regarding using terms such as mostly, often, frequently, only a minority, several, rarely... thank you for
your answer. However, in my opinion, the way it is written it does not give strength to what is presented, it is
far from precise and not easy to read”
We thank the reviewer for the comment. Even though our review was not systematic, we tried to be more specific
and precise in reporting the extracted data. Please see the “Evidence from the literature” paragraph.
